# Ulinastatin in the treatment of radiotherapy-induced oral mucositis in locoregionally advanced nasopharyngeal carcinoma: a phase 3 randomized clinical trial

Xuguang Wang[1,2,12], Haijun Wu[3,12], Feng Lei[4,12], Zhigang Liu [5], Guanzhu Shen[6], Xuefeng Hu[7], Yijing Ye[8], Manyi Zhu[9], Huageng Huang[1], Boyu Chen[10], Runda Huang[11], Chong Zhao [1,2,13] ✉, Jingjing Miao[1,2,13] ✉ & Lin Wang [1,2,13] ✉

Radiotherapy-induced oral mucositis (RTOM) is a common side effect of radiotherapy in locoregionally advanced nasopharyngeal carcinoma (LA-NPC) receiving concurrent chemoradiotherapy (CCRT). In this phase 3 trial, we aim to evaluate the efficacy and safety of Ulinastatin (UTI) for the prevention and treatment of RTOM in LA-NPC patients (NCT03387774). The primary endpoint is the incidence of grade ≥3 acute RTOM during radiotherapy. Secondary endpoints include cumulative incidence of RTOM, recovery rate, the onset time and duration of grade ≥3 RTOM, oral pain (severe), safety and survival outcomes. 179 eligible patients are randomly assigned to UTI Group ($n = 89$) or Control group ($n = 90$). All UTI group patients complete UTI treatment as planned, and both groups complete scheduled CCRT. The incidence of grade 3 RTOM is significantly lower in UTI group compared with control group (25.8% vs 41.1%, $P = 0.030$). The trial meet its prespecified primary endpoint. No Ulinastatin related adverse events are observed during treatment. The 3-year overall survival (OS), locoregional relapse-free survival (LRRFS), distant metastasis-free survival (DMFS) and progression-free survival (PFS) in UTI group and control group are similar between two groups. In this work, Ulinastatin can effectively reduce the severity of RTOM and oral pain without increasing toxicity and compromising survivals.

Nasopharyngeal carcinoma (NPC) is a type of head and neck tumor that occurs worldwide and has a geographic distribution, mainly in East and Southeast Asia[1]. The combination of radiotherapy and chemotherapy has become the establishd treatment for locoregionally advanced NPC (LA-NPC)[2]. A common side effect of radiotherapy in NPC is Radiotherapy-induced oral mucositis (RTOM), with an incidence of 50% ~ 100%[3,4]. Particularly, 30% to 50% of patients undergoing concurrent chemoradiotherapy experienced severe forms of

mucositis, classified as grade 3 or higher[5,6]. Severe RTOM can lead to significant discomfort for patients, including oral pain, difficulties in eating, an increase risk of malnutrition and treatment interruption, prolonged hospital stay, and a diminished effectiveness of the cancer treatment. Moreover, what starts as nonspecific inflammation can easily escalate to infectious inflammation, complicating treatment and leading to a higher reliance on antibiotics[7–11]. In recent 10 years, scholars have carried out a large number of studies on the prevention

A full list of affiliations appears at the end of the paper. ✉e-mail: gzzhaochong@hotmail.com; zhaochong@sysucc.org.cn; miaojingjing90@163.com; wangl1@sysucc.org.cn

and treatment of RTOM, such as the application of cell ischemia improving drugs, non-steroidal anti-inflammatory drugs, mucosal protective agents, and granulocyte-macrophage colony stimulating factors[12–15]. Despite these efforts, an effective prevention or treatment for RTOM remains elusive, with no medication yet recognized as a standard treatment for mucositis[16]. As such, symptomatic care, including local analgesics, remains the primary approach to management.

The main mechanisms of RTOM include: direct action of radiation on DNA of oral mucosal epithelial cells, local cell apoptosis, excessive activation of inflammatory pathways, and massive release of various inflammatory factors[17]. Ulinastatin is a refined glycoprotein extracted from human urine and is a protease inhibitor. It can inhibit the activity of trypsin and other pancreatic enzymes. Several cell experiments, animal experiments and clinical studies have shown that Ulinastatin can play an important role in suppressing inflammatory mediators by modulating inflammation related factors and regulating inflammatory response modulation signaling-related factors[18–21]. In the rat model of radiation-induced pulmonary injury, the expressions of TNF-a, TGF-β1 and IL-6 in the lung tissue of Ulinastatin treated rats were significantly decreased, and the degree of pneumonic exudation was reduced[18]. In our previous clinical practice, Ulinastatin was found to reduce the severity of inflammation in patients with RTOM, thereby relieving pain and improving treatment compliance, and no significant side effects were found. However, to our knowledge, the efficacy of Ulinastatin for patients with RTOM has not been prospectively investigated.

In this work, we present the results of an open-label, randomized, phase 3 trial evaluating the efficacy and safety of Ulinastatin for the prevention and treatment of RTOM in NPC patients.

## Results
### Patient and treatment characteristics
From 29 January 2018 to 28 December 2021, a total of 189 patients from five participating treatment centers were assessed for eligibility, and 179 (127 men [70.9%]; median age, 45 years [range, 36–52 years]) were enrolled in the trial and were randomly assigned to the UTI group ($n = 89$) or the control group ($n = 90$) (Fig. 1). Clinical characteristics were balanced between groups (Table 1[22]). All patients completed the pre-scribed course of IMRT, and no significant differences in radiotherapy parameters or treatment durations were observed. In the UTI group, 88 (98.9%) patients completed ≥2 cycles of concurrent cisplatin (1 received only 1 cycle due to grade 2 nephrototoxicity), while all patients completed ≥2 cycles of concurrent cisplatin in control group (Supplementary Table 2). During CCRT, all patients in UTI group received intravenous Ulinastatin at a dosage of 100,000 units three times daily.

### Efficacy
The incidence of grade 0–4 RTOM in two groups during CCRT seen Fig. 2A. None of the patients developed grade 4 RTOM. The incidence of grade 3 RTOM during CCRT in UTI group was significantly lower than that in control group (25.8% vs 41.1%, $P = 0.030$). From the beginning of CCRT to the 7th week, the cumulative rate of grade 3 RTOM in both groups gradually increased, especially in the control group (Fig. 2B). The onset times of grade 3 RTOM in UTI group and control group were 26.00 [19.00, 33.00] days and 32.00 [20.50, 36.00] days ($P = 0.621$), the duration of grade 3 RTOM from 0 to 7 weeks of radiotherapy in the two groups were 12.00 [7.00, 18.00] days and 15.00 [7.50, 25.50] days ($P = 0.393$), respectively (Supplementary Table 3, Supplementary Fig. 1). The rate of recovery from grade 3 RTOM (The proportion of patients with grade 3 RTOM who recovered to grade ≤2) during CCRT in UTI group was higher than that in control group (39.1% vs 10.8%, $P = 0.023$) (Supplementary Table 4). Eight patients (4 in the UTI group and 4 in the control group) interrupted radiotherapy because of holidays, and none of the two groups interrupted radiotherapy because of RTOM.

Patients in both groups developed mild, moderate, and severe oral pain during radiotherapy (Fig. 3A). Compared with the control group, the incidence of severe oral pain was significantly reduced in the UTI group (22.5% vs 36.7, $P = 0.038$). From the beginning of CCRT to the 7th week, the cumulative rate of severe oral pain in both groups gradually increased, especially in the control group (Fig. 3B). The rate of recovery from severe oral pain (The proportion of patients with severe oral pain who recovered to mild and moderate) during CCRT in UTI group was higher than that in control group (70.0% vs 48.5%, $P = 0.126$).

### Safety
During the entire treatment course, acute adverse events occurred in all patients (Table 2). The incidence of acute adverse events at all grades was similar between the two groups. Grade 3–4 acute adverse events were reported in 22 patients (24.7%) in the UTI group compared with 24 patients (26.7%) in the control group. The most frequent grade 3–4 acute adverse events were leukopenia (14 [15.7%] of 89 in UTI group versus 13 [14.4%] of 90 in the Control group), dry mouth (8 [8.9%] vs 8 [8.8%]), and neutropenia (4 [4.4%] vs 8 [8.8%]). 7 (7.9%) of 89 patients in the UTI group and 7 (7.8%) of 90 patients in the control group had late adverse events of grade 3 or 4 (Table 3). The most common grade 3 or worse late adverse event was auditory or hearing loss (5 [5.6%] of 89 patients in the UTI group vs 5 [5.6%] of 90 patients in the control group). There were no notable differences in late toxicities between the groups. No treatment-related deaths were reported.

### Follow-up and survival
At the last follow-up on 31 Jan 2024, median follow-up time for all 179 patients was 49.6 months (IQR, 36.8–60.9 months). We recorded 28 (15.6%) disease relapse or death events, including 11 (12.4%) in UTI group and 17 (18.9%) in control group. Details regarding the patterns of relapse and salvage therapies are shown in Supplementary Table 6 and 7. The 3-year OS, LRRFS, DMFS and PFS in the UTI group and Control group were 96.6% (95% confidence interval [CI], 93.0%–100.0%) and 94.4% (95% CI, 89.8%–99.3%) (HR 0.51, 95% CI 0.13–2.02, log-rank $p = 0.33$), 90.9% (95% CI, 85.1%–97.1%) and 87.7% (95% CI, 81.1%–94.8%) (HR 0.77, 95% CI 0.34–1.76, log-rank $p = 0.54$), 95.5% (95% CI, 91.3%–99.9%) and 91.1% (95% CI, 85.4%–97.2) (HR 0.37, 95% CI 0.12–1.16, log-rank $p = 0.08$), 89.8% (95% CI, 83.6%–96.3%) and 84.4% (95% CI, 77.2%–92.2%) (HR 0.64, 95% CI 0.30–1.37, log-rank $p = 0.17$), respectively (Supplementary Fig. 2).

## Discussion
This phase 3, multicenter, randomized clinical trial investigates the efficacy and safety of Ulinastatin in the prevention and treatment of RTOM in LA-NPC. The findings indicate that injections of Ulinastatin every weekday significantly reduce the incidence of grade ≥3 RTOM during radiotherapy.

Oral mucositis is a common radiation-induced toxicity in LA-NPC patients treated with CCRT. RTOM usually begins in the 2nd to 3rd week of radiotherapy and lasts for 4 to 8 weeks after treatment, depending on the radiotherapy technique, the radiation fractionation pattern, and the prevention strategies used[23,24]. In our study, the incidence of grade ≥3 RTOM in the control group was 41.1%, which is consistent with the results of previous studies[5,6]. RTOM can impair quality of life and interfere with treatment, or even cause patients to discontinue treatment entirely, thereby seriously compromising the effectiveness of antitumor therapy[25]. There is no standard treatment and it needs to be solved urgently. Previous studies have focused on locally applied agents, oral microbial load reduction agents, etc.[26], however, there are still no effective treatments, and new drugs need to be further explored and studied.

Ulinastatin is a kind of glycoprotein that can inhibit the activity of various proteolytic enzymes. It is a protease inhibitor and a natural

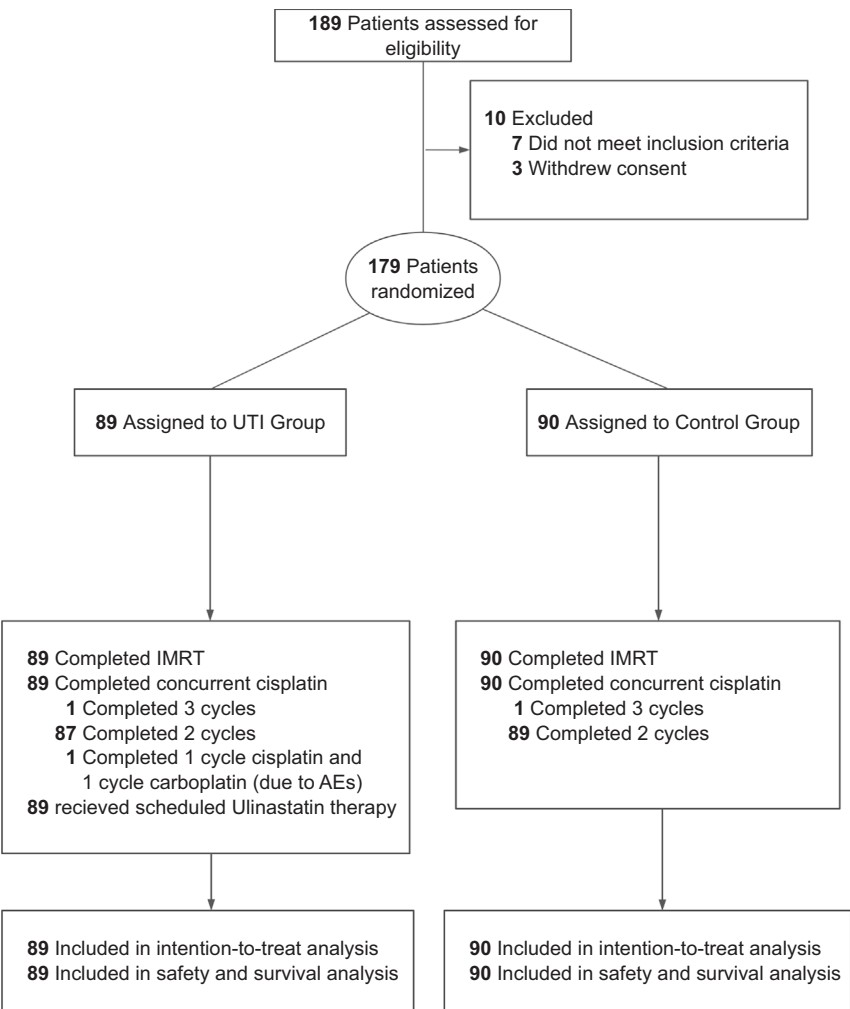

**Fig. 1 | Trial profile.** UTI Ulinastatin, CCRT concurrent chemoradiotherapy, AEs adverse events.

anti-inflammatory substance in human body. However, Ulinastatin content in the body is not enough to combat the inflammatory reaction in the body when the inflammatory reaction occurs, and it can be a broad-spectrum and effective at the same time to inhibit a variety of protease activity[27–29]. But the reports on the treatment of RTOM with Ulinastatin are relatively scanty. In our study, the incidence of grade ≥3 RTOM was 25.8% in UTI group and was significantly reduced by 15.3% compared to the control group, which is in line with our hypothesis. However, there was no statistical difference in the onset time and duration of grade ≥3 RTOM between the two groups. We only observed that the recovery rate in UTI group was higher than that in the control group from the beginning of radiotherapy to seventh week. This may be due to the fact that some patients with grade 3 RTOM in the control group persisted until some time after the end of radiotherapy, while some patients with grade 3 RTOM in the UTI group had recovered near the end of radiotherapy. During CCRT, the cumulative rate of grade ≥3 RTOM in two groups gradually increased, and a significant increase was observed from the 3rd week of radiotherapy, which also confirmed the results reported in other studies that RTOM usually begined in the 2nd to 3rd week of radiotherapy, but after that, compared with the control group, the overall increase trend of cumulative rate in UTI group was slower and continued to be lower. In this study, we observed that severe oral pain occurred in most patients with grade ≥3 RTOM, and the incidence of severe oral pain was significantly lower in UTI group. Ulinastatin appeared to reduce the incidence of severe pain by reducing the severity of oral mucositis.

At present, the optimal timing and dosage of Ulinastatin were unknown, and there are no studies on the comparison of treatment time and dose in the treatment of mucositis. According to the drug instructions, patients received intravenous Ulinastatin at a dosage of 100,000 units three times daily. The results demonstrate the good efficacy and safety of Ulinastatin, but intravenous injection three times a day may be inconvenient. Therefore, the optimal time and dosage of the drug still need to be further explored. It was reported that the incidence of ADRs/ADEs of Ulinastatin is < 5‰. The ADRs/ADEs involved limited organs, mainly the skin, gastrointestinal tract, and blood. The higher the dose of Ulinastatin, the higher the incidence of ADRs/ADEs will generally be. In most cases, the ADRs/ADEs gradually alleviated or recovered after drug withdrawal[30]. Up to now, several clinical studies have shown that the daily therapeutic dose of Ulinastatin can reach 1,200,000–5,000,000 units, which also demonstrated a favorable safety profile with high doses of Ulinastatin[31,32]. The dose of the drug in our study was lower than those in other studies, so the results also showed the use of Ulinastatin is safer.

So far, there is no evidence from animal and clinical studies that Ulinastatin promotes the growth of any tumor cells. In our study, the overall survival outcomes were slightly better in the UTI group than that in the control group, but did not show a significant difference. Recently, by treating highly-metastatic NPC cell lines S18 and 58 F with UTI, Li et al. found that UTI suppressed the migration and infiltration of S18 and 5–8 F cells and suppressed the metastasis of S18 cells in vivo by downregulating the expression of uPA and uPAR, thus partially

**Table 1 | Clinical characteristics of parents**

| Baseline characteristics | Group, no. (%) UTI group (n = 89) | Control Group (n = 90) |
|---|---|---|
| Age, median (IQR), y | 43 (35–51) | 46 (37–52) |
| Sex | | |
| Male | 60 (67.4) | 67 (74.4) |
| Female | 29 (32.6) | 23 (25.6) |
| Underlying disease | | |
| Yes | 35 (39.3) | 26 (67.4) |
| No | 54 (60.7) | 64 (28.9) |
| Smoking status | | |
| Yes | 24 (27.0) | 36 (40.0) |
| No | 65 (73.0) | 54 (60.0) |
| Family history | | |
| Yes | 23 (25.8) | 18 (20.0) |
| No | 66 (74.2) | 72 (80.0) |
| KPS score | | |
| >90 | 4 (4.5) | 3 (3.3) |
| ≤90 | 85 (95.5) | 87 (96.7) |
| T stage[a] | | |
| T1 | 4 (4.5) | 1 (1.1) |
| T2 | 4 (4.5) | 4 (4.4) |
| T3 | 67 (75.3) | 67 (74.4) |
| T4 | 14 (15.7) | 18 (20.0) |
| N stage[a] | | |
| N0 | 11 (12.4) | 11 (12.2) |
| N1 | 50 (56.2) | 40 (44.9) |
| N2 | 23 (25.8) | 33 (36.7) |
| N3 | 5 (5.6) | 6 (6.7) |
| TNM stage[a] | | |
| III | 71 (79.8) | 68 (75.6) |
| IV | 18 (20.2) | 22 (24.4) |
| Pre-treatment EBV-DNA copies | | |
| >500 | 30 (33.7) | 28 (31.1) |
| ≤500 | 59 (66.3) | 62 (68.9) |

*UTI* Ulinastatin, *EBV* Epstein-Barr virus, *KPS* Karnofsky performance status.
[a]Histologic characteristics were categorized according to the World Health Organization classification of tumors[22].

inhibiting the metastasis of nasopharyngeal carcinoma[33]. Therefore, it is possible that significant differences may be observed over longer follow-up, especially in DMFS between the two groups. These present results suggested that the use of Ulinastatin was unlikely to compromise the efficacy of CCRT in patients with LA-NPC, nor did it affect the survival outcomes of patients.

The study had several limitations: First, the optimal time and dosage of Ulinastatin is still uncertain and requires further research. At the same time, intravenous injection three times a day is not convenient for the clinical management of patients, so it is necessary to explore better medication modes. Seocndly, less data were collected on patients' quality of life during and after treatment, thereby failing to assess the overall health of patients in multiple ways. In addition, longer-term follow-up is needed to analyze the effect of Ulinastatin on survival outcomes. The lack of blinding is another weakness of the study.

In conclusion, this phase 3 randomized clinical trial shows that Ulinastatin is effective and safe in the prevention and treatment of oral mucositis in patients with LA-NPC. It can significantly reduce the severity of RTOM and oral pain without increasing toxicity and compromising survivals, and also delay the onset of severe toxicity.

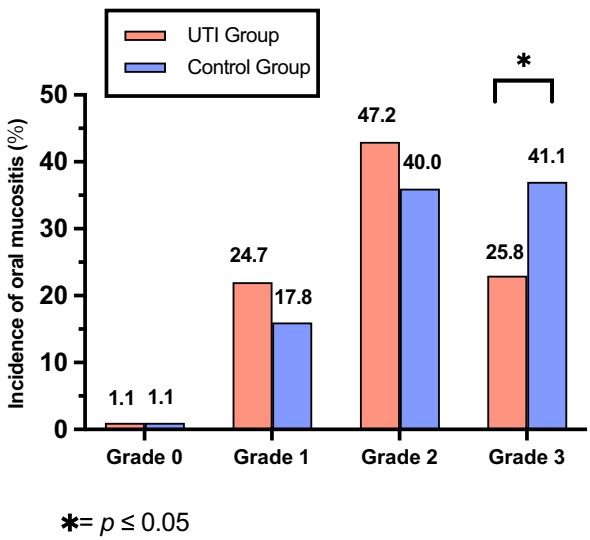

(A)

**∗**= *p* ≤ 0.05

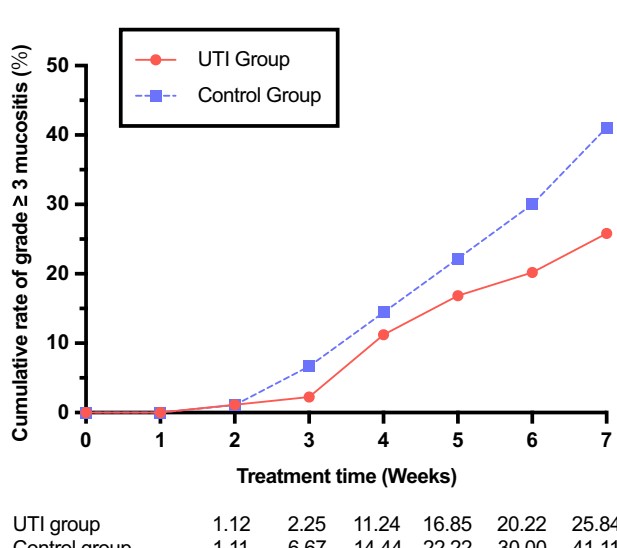

(B)

| | | | | | | |
|---|---|---|---|---|---|---|
| UTI group | 1.12 | 2.25 | 11.24 | 16.85 | 20.22 | 25.84 |
| Control group | 1.11 | 6.67 | 14.44 | 22.22 | 30.00 | 41.11 |
| *P* value | 1.000 | 0.285 | 0.521 | 0.365 | 0.132 | 0.030 |

**Fig. 2 | Acute oral mucosities by treatment groups. A** Incidence of acute oral mucosities for 179 evaluable patients. The difference in incidence of grade 3 RTOM during CCRT between groups was statistically significant (*P* = 0.030, two-sided Fisher's exact test). Statistical note: no adjustments for multiple comparisons were applied, as the analysis focused on pre-specified differences in grade 3 RTOM. **B** Cumulative rate of grade ≥3 mucositis during treatment in UTI group (89 patients, at 7 weeks 23 at risk), Control Group (90 patients, at 7 weeks 37 at risk). Source data are provided as a Source Data file. *: Significant difference between groups (two-sided Fisher's exact test; *P* < 0.05). UTI Ulinastatin.

## Methods
### Study oversight
The protocol was approved by the Research Ethics Board of Sun Yat-sen University Cancer Center (SYSUCC) and was performed in accordance with the Declaration of Helsinki. The study design and conduct complied with all relevant regulations regarding the use of human study participants. All patients provided written informed consent. The study was registered on ClinicalTrials.gov with the identifier NCT03387774.

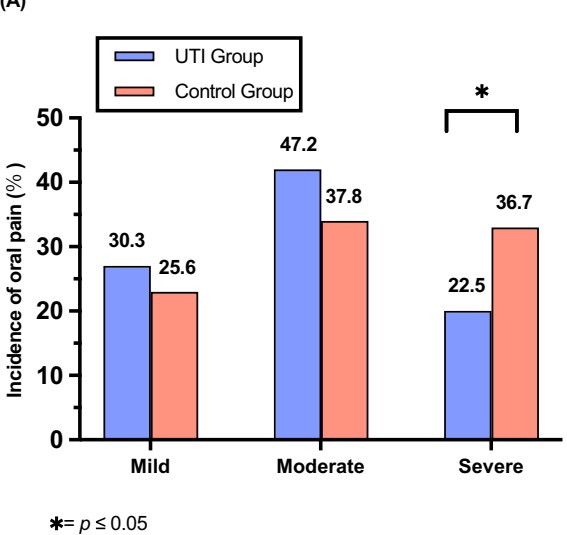

**(A)**

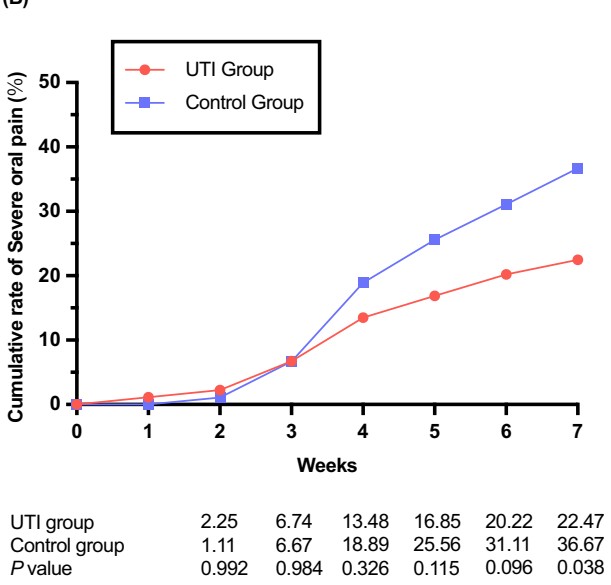

**(B)**

| | | | | | | |
|---|---|---|---|---|---|---|
| UTI group | 2.25 | 6.74 | 13.48 | 16.85 | 20.22 | 22.47 |
| Control group | 1.11 | 6.67 | 18.89 | 25.56 | 31.11 | 36.67 |
| *P* value | 0.992 | 0.984 | 0.326 | 0.115 | 0.096 | 0.038 |

**Fig. 3 | Oral pain by treatment groups. A** Incidence of oral pain for 179 evaluable patients. The difference in the incidence of severe oral pain during CCRT between groups was statistically significant ($P = 0.038$, two-sided Fisher's exact test). Statistical note: no adjustments for multiple comparisons were applied, as the analysis focused on pre-specified differences in severe oral pain. **B** Cumulative rate of severe oral pain during treatment in UTI group (89 patients, at 7 weeks 20 at risk), Control group (90 patients, at 7 weeks 33 at risk). Source data are provided as a Source Data file. *: Significant difference between groups (two-sided Fisher's exact test; $P < 0.05$). UTI Ulinastatin.

## Study design and participants

This multi-center, open, randomized controlled clinical trial was conducted at 5 hospitals in China. Patients who fulfilled the following criteria were eligible: 1) newly diagnosed, pathologically confirmed NPC, with no distant metastasis; 2) stage III or stage IVa according to the staging system of the 8th American Joint Committee on Cancer/ Union for International Cancer Control (AJCC/UICC); 3) aged between 18 and 65 years; 4) adequate organ function; 5) a Karnofsky

performance Status (KPS score) ≥80. The exclusion criteria were as follows: 1) Prior anti-tumor therapy; 2) metachronous or synchronous malignancy; 3) pregnant or lactating female; 4) pre-existing oral mucositis; 5) unwilling to discontinue harmful habits such as smoking, drinking alcohol and chewing betel nut; 6) severe comorbidities.

## Randomization and masking

Simple Random Sampling (SRS) was used and randomization was performed at SYSUCC utilizing a computer-generated sequence to obtain the randomization list without stratification. Patients were randomly assigned before starting treatment to receive either Ulinastatin plus CCRT (UTI group) or CCRT alone (control group). After obtaining informed consent, the investigators at each institution contacted the study coordinators at the Clinical Trials Center and received treatment assignment information. The statistician (Y.G.) and study coordinators were uninvolved in the treatment of patients and data monitoring. The treatment assignment was unmasked to both patients and clinicians. The Investigators were not blinded to allocation during experiments and outcome assessment.

## Procedures

All of the patients underwent CCRT, and the chemotherapy regimen contained 100 mg/m² cisplatin every 3 weeks for 2 to 3 cycles, depending on the duration of radiotherapy. No patient received induction chemotherapy. Chemotherapy dose modifications were based on the nadir blood counts and acute toxic effects of the preceding cycle. The standard for chemotherapy in participating treatment centers is that all patients have an indwelling Peripherally Inserted Central Catheter (PICC). The UTI group was treated with Ulinastatin through intravenous drip at a dosage of 100,000 units added to 100 ml of 0.9% normal saline, 3 times every radiation day, until the end of radiotherapy. Ulinastatin was administered at 8 a.m., 2 p.m., and 8 p.m, with 6-hour intervals between doses. Except for the use of Ulinastatin, RTOM was managed similarly in both groups (Supplementary information). This included a stepwise approach to cancer pain, providing nutritional support, and administering antibiotics in cases of coinfection. Patients were encouraged to supplement their nutrition with oral nutritional supplements (ONS) under adequate pain management. For those who cannot meet their nutritional needs through ONS, the feasibility of nasogastric tube placement or gastrostomy we assessed. If patients decline these options, parenteral nutrition is considered. Glutamine and light therapy were not used in this trial. The criteria for discontinuing Ulinastatin were as follows: 1. The patient refused Ulinastatin treatment; 2. Allergic reaction during treatment; 3. Intolerance of associated toxicity.

Patients in both groups were treated with radical intensity-modulated radiation therapy (IMRT). All target volumes and organ at risk (OAR) were outlined slice by slice on the axial contrast-enhanced CT with MR fusion images in the treatment planning system of Eclipse. The target volumes were defined in accordance with the International Commission on Radiation Units, and Measurements Reports 50 and 62. The prescribed dose was 68–72 Gy to PTVnx (Planning target volume of the primary tumor), 64–68 Gy to GTVnd (Gross tumor volume of the cervical lymph node), 60–64 Gy to PTVnd and PTV1 (Planning target volume 1), and 54–58 Gy to PTV2 (Planning target volume 2) in 30–32 fractions[34]. The details of dose limits for OAR were based on the study 0225 from The Radiation Therapy Oncology Group (RTOG 0225)[35].

All patients were followed up at 1 and 3 months after the end of IMRT, then every 3 months for the first 3 years, every 6 months for the next 2 years, and annually thereafter. Hematologic and biochemical analyses, nasopharyngoscopy and contrast-enhanced MRI were performed at each follow-up. If tumor recurrence or metastasis is diagnosed, further treatment is determined on a case-by-case basis.

**Table 2 | Treatment-related acute adverse events**

| | Patients, no. (%) UTI group (n = 89) | | | | Control group (n = 90) | | | |
| | Grade 1 | Grade 2 | Grade 3 | Grade 4 | Grade 1 | Grade 2 | Grade 3 | Grade 4 |
|---|---|---|---|---|---|---|---|---|
| Any acute adverse event[a] | 9 (10.1) | 58 (65.2) | 21 (23.6) | 1 (1.1) | 8 (8.9) | 58 (64.4) | 24 (26.7) | **0** |
| Hematologic toxic effects | | | | | | | | |
| Leukopenia | 15 (16.8) | 43 (48.3) | 13 (14.6) | 1 (1.1) | 14 (15.5) | 45 (50.0) | 13 (14.4) | 0 |
| Neutropenia | 19 (21.3) | 21 (23.5) | 4 (4.4) | 0 | 20 (22.2) | 13 (14.4) | 8 (8.8) | 0 |
| Anemia | 29 (32.5) | 9 (10.1) | 2 (2.2) | 0 | 20 (22.2) | 6 (6.7) | 0 | 0 |
| Thrombocytopenia | 7 (7.8) | 3 (3.4) | 1 (1.1) | 0 | 7 (7.8) | 1 (1.1) | 1 (1.1) | 0 |
| Nonhematologic toxic effects | | | | | | | | |
| Nausea and vomiting | 77 (86.5) | 12 (13.4) | 0 | 0 | 76 (84.4) | 13 (14.4) | 1 (1.1) | 0 |
| Hepatotoxicity | 8 (8.9) | 2 (2.2) | 1 (1.1) | 0 | 7 (7.8) | 4 (4.4) | 1 (1.1) | 0 |
| Nephrotoxicity | 2 (2.2) | 3 (3.4) | 0 | 0 | 5 (5.6) | 0 | 0 | 0 |
| Dry mouth | 32 (35.9) | 45 (50.5) | 8 (8.9) | 0 | 31 (34.4) | 46 (51.1) | 8 (8.9) | 0 |
| Auditory/hearing | 32 (35.9) | 25 (28.0) | 1 (1.1) | 0 | 37 (41.1) | 19 (21.1) | 6 (6.7) | 0 |
| Skin/neck tissue damage | 53 (59.5) | 8 (9.0) | 0 | 0 | 48 (53.3) | 3 (3.3) | 0 | 0 |

*UTI* ulinastatin.
[a]Acute adverse events were graded according to the Common Terminology Criteria for Adverse Events (version 5.0).

**Table 3 | Treatment-related late adverse events**

| | Patients, no. (%) UTI group (n = 89) | | | | Control Group (n = 90) | | | |
| | Grade 1 | Grade 2 | Grade 3 | Grade 4 | Grade 1 | Grade 2 | Grade 3 | Grade 4 |
|---|---|---|---|---|---|---|---|---|
| Any late adverse events[a] | 48 (53.9) | 15 (16.9) | 7 (7.9) | 0 | 49 (54.4) | 10 (11.1) | 7 (7.8) | 0 |
| Dry mouth | 44 (49.4) | 6 (6.7) | 1 (1.1) | 0 | 45 (50.0) | 4 (4.4) | 1 (1.1) | 0 |
| Auditory/hearing loss | 22 (24.7) | 11 (12.4) | 5 (5.6) | 0 | 30 (33.3) | 6 (6.7) | 5 (5.6) | 0 |
| Peripheral neuropathy | 11 (12.4) | 0 | 0 | 0 | 17 (18.9) | 0 | 0 | 0 |
| Skin fibrosis | 22 (24.7) | 0 | 0 | 0 | 14 (15.6) | 0 | 0 | 0 |
| Dysphagia | 8 (9.0) | 0 | 0 | 0 | 15 (16.7) | 1 (1.1) | 0 | 0 |
| Eye damage | 1 (1.1) | 0 | 1 (1.1) | 0 | 1 (1.1) | 0 | 1 (1.1) | 0 |
| brain injury | 1 (1.1) | 0 | 0 | 0 | 7 (7.7) | 0 | 0 | 0 |

*UTI* ulinastatin.
[a]Late adverse events were graded according to the Late Radiation Morbidity Scoring Scheme of the Radiation Therapy Oncology Group.

## Outcomes

Primary endpoint of this study was the incidence of graded ≥3 acute RTOM during radiotherapy in both groups (the proportion of all patients with grade ≥3 acute RTOM from 0 to 7 weeks of radiotherapy to the total number of patients observed). Secondary endpoints included the following: 1. cumulative rate of grade ≥3 RTOM from 0 to 7 weeks of radiotherapy (the proportion of patients who developed grade ≥3 acute RTOM from the start of radiotherapy to the end of the observed week, out of the total number of patients); 2. onset time of grade ≥3 RTOM ; 3. duration of grade ≥3 RTOM from 0 to 7 weeks of radiotherapy ; 4. recovery rate (proportion of patients with grade ≥3 RTOM who recovered to grade ≤ 2 during CCRT); 5. severe oral pain; 6. safety; 7. completion rate of planned CCRT and the incidence of radiotherapy interruption (radiotherapy interruption ≥5 days); 8. overall survival (OS, durations were calculated from the date of randomization to the date of last follow-up or death from any cause), locoregional relapse-free survival (LRRFS, durations were calculated from the date of randomisation and the date of locoregional recurrence, or death from any cause), distant metastasis-free survival (DMFS, durations were calculated from the date of randomisation and the date of distant metastasis, or death from any cause) and progression-free survival (PFS, durations were calculated from the date of randomisation to the date of locoregional recurrence, distant

metastasis, or death from any cause, whichever occurred first). Patients unavailable for follow-up or alive without distant metastasis or locoregional relapse were censored at the date of last follow-up.

The grade of RTOM and oral pain were recorded every week from the start of radiotherapy for each patient until the end of the seventh week. The acute toxicity criteria of the Radiation Therapy Oncology Group (RTOG) (scored by the physician) was used for RTOM assessment. Oral pain was assessed according to the numerical rating scale (NRS) pain scores. RTOM is classified as grade 0–4 according to the acute toxicities criteria of the RTOG: grade 0 means an absence of any oral discomfort; grade 1 means erythema mucous membrane in the oral cavity; grade 2 means the presence of spotty ulcers or scattered mucosal leukoplakia in the oral cavity; grade 3 means fused ulcers, fused mucosal white spots and bleeding due to minor trauma in the oral cavity; grade 4 means tissue necrosis or significant spontaneous bleeding in the oral cavity or life-threatening complications. NRS was also known as the pain intensity numerical rating scale (PINRS); the method was to ask the patient to describe their pain level using 11 numbers from 0 to 10. The grading criteria of NRS were as follows: 0 = no pain, 1–3 = mild pain, 4–6 = moderate pain, 7–10 = severe pain.

Acute chemotherapy toxic reactions during treatment were graded according to the Common Terminology Criteria for Adverse Events, version 5.0 (CTCAE v5.0). Radiotherapy toxicity was assessed

according to the radiation damage grading criteria proposed by the RTOG[36]. The adverse drug reactions/adverse drug events (ADRs/ADEs) of Ulinastatin including the skin, gastrointestinal tract and blood, were recorded separately in the study[30].

## Statistical analysis

In this study, The total sample size was calculated using the Power and Sample Size Calculation (PASS), version 14.0, software[37]. According to previous studies, the incidence of grade ≥3 oral mucositis in NPC treated by IMRT with cisplatin was about 40%[5,38]. Assuming that the incidence of grade ≥3 RTOM in the Ulinastatin group decreased to 20%, the significance test level $\alpha = 0.05$, Power = 0.8, at least 80 subjects in each group were estimated. At least 176 subjects (88 per group) were required to be enrolled in this study based on a 10% dropout and loss of follow-up rate. No data were excluded from the analyses. All statistical analyses were performed using the SPSS software, version 25.0 (IBM Corp., Chicago, IL, USA) and were 2-sided at a significance level of $P < 0.05$. Categorical variables were compared using the $\chi^2$ test or the Fisher exact test. Continuous variables were compared using the Mann–Whitney test or Two Independent Sample T-Test. Survival curves were estimated using the Kaplan-Meier method and were compared by the log-rank test.

## Reporting summary

Further information on research design is available in the Nature Portfolio Reporting Summary linked to this article.

## Data availability

The data related to oral mucositis and oral pain generated in this study have been provided in the Source Data file. The Clinical raw data are not publicly available due to involving patient privacy, but can be accessed on request from the corresponding author for 10 years. Any request should be sent to the corresponding author, L.W., via email at wangl1@sysucc.org.cn, along with a detailed description of your research protocol; individual deidentified participant data will be shared. Please allow one month for response to requests. The corresponding author and Sun Yat-sen University Cancer Center will evaluate the reasonability of the request for our data and reserve the right to decide whether to share the data or not. And the data is only used for the research purpose. All data shared will be de-identified and will be available for 1 year after access is granted. The study protocol is available as Supplementary Note in the Supplementary Information file. The CONSORT checklist is also available in the Supplementary Information file. Source data are provided with this paper.

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

## Acknowledgements

This work was supported by the National Natural Science Foundation of China [No. 82073330], National Natural Science Foundation of China for Young Scholars [No. 82202946], Sun Yat-sen University Cancer Center 308 Program [308-2016-04-01], and Immune-radiotherapy Research Foundation of Chinese Medical Association Radiation Oncology [Z-2017-24-2108]. We thank the patients who participated in this study, their families, and the medical, nursing, and research staff at the study center. The sponsor(s) had no role in study design, data collection and analysis, or manuscript writing.

## Author contributions

L.W., J.J.M., C.Z. conceived and designed the study. All authors (X.G.W., H.J.W., F.L., Z.G.L., G.Z.S., X.F.H., Y.J.Y., M.Y.Z., H.G.H., B.Y.C., R.D.H., C.Z., J.J.M., L.W.) performed patient recruitment and data acquisition. X.G.W. performed the statistical analysis, had full access to all the data in the study, and took responsibility for the integrity of the data and the accuracy of the data analysis. L.W., J.J.M., C.Z. obtained funding. L.W., J.J.M., C.Z., F.L., Z.G.L., G.Z.S., X.F.H., Y.J.Y. provided administrative, technical or material support. C.Z., L.W. conducted the research supervision. X.G.W., J.J.M., L.W., C.Z. drafted the manuscript. All authors approved the final manuscript and agree to be accountable for all aspects of the work.

## Competing interests

The authors declare no competing interests.

### Ethics approval

The study was approved by the Ethics Committee of Sun Yat-sen University Cancer Center (B2017-063-01-01) and registered on ClinicalTrials.gov, number NCT03387774.

### Additional information

¹Sun Yat-sen University Cancer Center, State Key Laboratory of Oncology in South China, Collaborative Innovation Center for Cancer Medicine, Guangdong Key Laboratory of Nasopharyngeal Carcinoma Diagnosis and Therapy, Guangzhou, People's Republic of China. ²Department of Nasopharyngeal Carcinoma, Sun Yat-sen University Cancer Center, Guangzhou, People's Republic of China. ³Department of Comprehensive (Head and Neck) Oncology and Hospice Ward, First People's Hospital of Foshan, Foshan, China. ⁴Head and Neck Radiotherapy Department, The People's Hospital of Zhongshan City, Zhongshan, Guangdong, China. ⁵Cancer Center, the 10th Affiliated Hospital of Southern Medical University (Dongguan People's Hospital), Southern Medical University, Guangzhou, China. ⁶Department of Radiation Oncology, The Third Affiliated Hospital of Sun Yat-sen University, Guangzhou, China. ⁷Department of Radiation Oncology, First People's Hospital of FoShan Affiliated with Sun Yat-Sen University, Foshan, China. ⁸Department of Abdominal Tumor Radiotherapy, Zhongshan City People's Hospital, Zhongshan, Guangdong, China. ⁹Department of Radiation Oncology, National Cancer Center/National Clinical Research

Center for Cancer/Cancer Hospital and Shenzhen Hospital, Chinese Academy of Medical Sciences and Peking Union Medical College, Shenzhen, Guangdong, PR China. [10]Department of Experimental Research, State Key Laboratory of Oncology in South China, Collaborative Innovation Center for Cancer Medicine, Sun Yat-sen University Cancer Center, Guangzhou, China. [11]Department of Radiation Oncology, Sun Yat-sen University Cancer Center, Guangzhou, People's Republic of China. [12]These authors contributed equally: Xuguang Wang, Haijun Wu, Feng Lei. [13]These authors jointly supervised this work: Chong Zhao, Jingjing Miao, Lin Wang. The results of this study were presented in a Poster Session (Head and Neck Cancer) at 2024 Annual Meeting of American Society of Clinical Oncology (ASCO), June 2, 2024. ✉e-mail: gzzhaochong@hotmail.com; zhaochong@sysucc.org.cn; miaojingjing90@163.com; wangl1@sysucc.org.cn

