## [Transparent Peer Review file · Nature Communications]

Ulinastatin in the treatment of radiotherapy-induced oral mucositis in locoregionally advanced nasopharyngeal carcinoma : A Phase 3 Randomized Clinical Trial

Corresponding Author: Dr Lin Wang

Version 0:

Reviewer comments:

Reviewer #1

(Remarks to the Author)

Congratulations on this trial, which addresses an important topic in radiotherapy.

1. The primary outcome is stated to be "... the incidence of graded [sic] ≥ 3 acute RTOM in 392 both groups." Were RTOMs recorded during treatment only or also observed a period of time after treatment - and, if so, how long?
2. Likewise, how long were subjects followed for OS and other survival endpoints?
3. Please define "cumulative incidence" of grade ≥ 3 toxicities. How does it differ from incidence?
4. I have never heard of the "power analysis and sample size" software and no citation is given.
5. Lines 440, 441 refer to "cases", whose most logical meaning would refer to RTOM, of course not the meaning here. Please substitute "subjects".
6. Overall incidence of acute toxicities is about the same in the two groups, as is incidence of pain. The apparent effect of the treatment is to reduce each endpoint's severity rather than reduce their incidence. The results should be presented in this manner.
7. The last sentence of the Results section states that "although it does not delay the onset of grade 3 toxicity." Figure 3 appears to show that it does delay onset.
8. Details of the randomization algorithm should be given.
9. The discussion should state that a weakness of this study is its lack of blinding.
10. Since, judging from the supplementary figures on survival, you have at least two years' followup on all subjects with some extending to 6 years, you should be able to report results on late toxicities. They would be useful.

Rick Chappell
Department of Biostatistics and Medical Informatics
The University of Wisconsin
chappell@stat.wisc.edu

Reviewer #2

(Remarks to the Author)

The authors present results from a multicenter, open-label, randomized-controlled trial conducted at five hospital in China. The authors hypothesized that the protease inhibitor ulinastatin would decrease the incidence of grade 3-5 radiotherapy-

associated oral mucositis (RTOM).

My comments are as follows:

1. This trial enrolled patients with AJCC 8th edition stage III-IVA NPC from 2018-2021. Did patients receive induction chemotherapy? If so, please list the regimens and assess for balance between cohorts. Induction chemotherapy may result in OM, and pre-existing OM was an exclusion criterion for this trial.
2. Page 17: Can the authors confirm that ulinastatin was given three times daily via IV on every day of radiotherapy? How was this logistically completed? Did patients have indwelling ports or PICCs? What interval separated ulinastatin administration? What were criteria for discontinuation of ulinastatin? Can the authors please describe previously-reported potential side effects or toxicities of ulinastatin in the introduction or discussion?
3. Page 17: Please expand upon the standard approach for managing RTOM. What standard interventions did these five centers typically use? What types of oral analgesics were typically used? What types of oral and/or nasal rinses were typically used? Were any other mucositis-modifying agents used (e.g., glutamine, light therapy, etc). What were typical criteria for feeding tube placement? Please describe the general approach to treating presumed infections with antibiotics, as need for antibiotics is not commonly observed when treating oropharyngeal carcinomas with cisplatin-based chemoradiotherapy.
4. "RTOM can disrupt therapy and even drive patients to stop it altogether. It can also hasten the growth of remaining tumor cells, trigger metastasis or recurrence of the original tumor, and lower patients' long-term survival rates." These statements are not fully supported by high-level clinical evidence. RTOM certainly impairs quality of life and may lead to interruptions/discontinuation of therapy, but a direct link between RTOM and tumor growth is not established.

Version 1:

Reviewer comments:

Reviewer #1

(Remarks to the Author)

1. You successfully addressed all my suggestions. I was especially pleased that you were able to add information on late effects. Congratulations.
2. Your response to my earlier comment: "3. Please define "cumulative incidence" of grade ≥ 3 toxicities. How does it differ from incidence?" implied that cumulative rate (as you now label it) at 7 weeks should equal incidence. In Figure 2 this is true for the control but not for the UTI group (26.97 vs. 25.8). Why are these figures different?

Rick Chappell
chappell@stat.wisc.edu

Reviewer #2

(Remarks to the Author)

1. In response to Reviewer #2 Comment #1, I recommend explicitly stating on page 18 that no patient received induction chemotherapy.
2. In response to Reviewer #2 Comment #2, I recommend explicitly stating in the manuscript that a) patients received IV Ulinastatin at 8am, 2pm, and 8pm on every day of radiotherapy, b) the standard for chemotherapy in this region is that all patients have an indwelling PICC, and c) the criteria for discontinuation of Ulinastatin as stated by the authors in their response.
3. In response to Reviewer #2 Comment #3, it is stated that intravenous nutrition was administered. Was enteral nutrition (nasogastric, gastrostomy) not performed for mucositis limiting oral intake? I recommend explicitly stating that glutamine and light therapy were not used in this trial.

Authors' response

Reviewer #1

Reviewer's comments	Authors' responses or change made	Page number
Congratulations on this trial, which addresses an important topic in radiotherapy.	Thank you very much for the positive comment.	
1. The primary outcome is stated to be "... the incidence of graded [sic] ≥ 3 acute RTOM in 392 both groups." Were RTOMs recorded during treatment only or also observed a period of time after treatment - and, if so, how long?	We thank the reviewer for the important feedback. In this study, RTOM were recorded only during radiotherapy, weekly from the start of radiotherapy for each patient until the end of the seventh week, as mentioned in line 411-412. We have revised the wording to " Primary endpoint of this study was the incidence of graded ≥ 3 acute RTOM during radiotherapy in both groups (the proportion of all patients with grade ≥ 3 acute RTOM from 0 to 7 weeks of radiotherapy to the total number of patients observed)".	Revised lines 416-419 on page 19-20
2. Likewise, how long were subjects followed for OS and other survival endpoints?	Many thanks for your reminder. We conducted long-term follow-up for OS and other survival endpoints of all patients until Jan. 31, 2024, and the median follow-up time was 49.6 months. Accordingly, we have revised it as "At the last follow-up on Jan. 31, 2024, median follow-up time for all 179 patients was 49.6 months (IQR, 36.8-60.9 months)."	Revised line 214 on page 10
3. Please define "cumulative incidence" of grade ≥ 3 toxicities. How does it differ from incidence?	Thank you for your insightful comment. We are sorry for the ambiguity in our initial expression which may have caused confusion. We have replaced "cumulative incidence" with "cumulative rate", and define both as follows: 1.Incidence: In the manuscript, "incidence of graded ≥ 3 acute RTOM" refers to the proportion of patients with graded ≥ 3 acute RTOM to the total number of patients observed during radiotherapy. 2.Cumulative rate: In the manuscript, "cumulative rate of grade ≥ 3 RTOM from 0 to 7 weeks of radiotherapy" refers to the proportion of patients who developed grade ≥ 3 acute RTOM from the start of radiotherapy to the end of the observed week, out of the total number of patients.We have also added the	Revised lines 416-422 on 19-20

	definitions of both in the revised manuscript.	
4. I have never heard of the "power analysis and sample size" software and no citation is given.	Thanks for your comment. Power and sample size calculation (PASS) is a statistical tool used to determine the required sample size or to analyze the potency of a study, and is widely used in clinical trials of nasopharyngeal carcinoma. ¹⁻³ According to your suggestions, we have revised as follows: "In this study, The total sample size was calculated using the Power and Sample Size Calculation (PASS) , version 14.0, software." ⁴	Revised lines 464-465 on page 22
5. Lines 440, 441 refer to "cases", whose most logical meaning would refer to RTOM, of course not the meaning here. Please substitute "subjects".	Many thanks for your suggestion. According to your suggestion, we have replaced "cases" with "subjects" in the revised version.	Revised lines 469-471 on page 22
6. Overall incidence of acute toxicities is about the same in the two groups, as is incidence of pain. The apparent effect of the treatment is to reduce each endpoint's severity rather than reduce their incidence. The results should be presented in this manner.	Thank you for your valuable suggestion. In this study, the reduced incidence of grade 3 or higher RTOM indicates a decrease in the severity of RTOM. Likewise, the decreased incidence of severe oral pain (NRS score ≥ 7) indicates a reduction in the severity of oral pain. According to the Reviewer's suggestion, we revised the conclusions as follows, "Ulinastatin can effectively reduce the severity of RTOM and oral pain without increasing toxicity and compromising survivals."	Revised lines 105-107 on page 5

7. The last sentence of the Results section states that "although it does not delay the onset of grade 3 toxicity." Figure 3 appears to show that it does delay onset.	We greatly appreciate your helpful reminder. Combined with the results of this study, we consider that Ulinastatin can reduce the severity of RTOM and severe oral pain, but also delay the onset of toxicity. Therefore, we have revised the sentence as follows, "It can significantly reduce the severity of RTOM and oral pain without increasing toxicity and compromising survivals, and also delay the onset of severe toxicity."	Revised lines 334-336 on page 16
8. Details of the randomization algorithm should be given.	We appreciate your professional suggestion on this matter. In this study, the disease types, stages, and treatment of the patients were relatively uniform, with few confounding factors. The comparison of baseline data also showed that the two groups were balanced. Therefore, Simple Random Sampling and non-stratified method were used. The content of randomization has been modified and supplemented in the revised manuscript, which is as follows: Simple Random Sampling (SRS) was used and randomization was performed at SYSUCC utilizing a computer-generated sequence to obtain the randomization list without stratification.	Revised lines 372-374 on page 17
9. The discussion should state that a weakness of this study is its lack of blinding.	Many thanks for your valuable suggestion. We added the following to the discussion section of the revised manuscript: "The lack of blinding is another weakness of the study."	Revised lines 329-330 on page 15

10. Since, judging from the supplementary figures on survival, you have at least two years' follow up on all subjects with some extending to 6 years, you should be able to report results on late toxicities. They would be useful.

We thank for this constructive comment. We have compiled the patient's late toxicity data, which is summarized in the table below and Table 2 in the revised manuscript. Also, the results of late toxicity has been added to the "Results" section of the manuscript.

Revised lines 199-211 on page 10 and Table 2

Treatment-Related Adverse Events

	Patients, No. (%)							
	UTI Group (n=89)				Control Group (n=90)			
	Grade 1	Grade 2	Grade 3	Grade 4	Grade 1	Grade 2	Grade 3	Grade 4
Any acute adverse event*	9 (10.1)	58 (65.2)	21 (23.6)	1 (1.1)	8 (8.9)	58 (64.4)	24 (26.7)	0
Hematologic toxic effects								
Leukopenia	15 (16.9)	43 (48.3)	13 (14.6)	1 (1.1)	14 (15.6)	45 (50.0)	13 (14.4)	0
Neutropenia	19 (21.3)	21 (23.6)	4 (4.5)	0	20 (22.2)	13 (14.4)	8 (8.9)	0
Anemia	29 (32.6)	9 (10.1)	2 (2.2)	0	20 (22.2)	6 (6.7)	0	0
Thrombocytopenia	7 (7.9)	3 (3.4)	1 (1.1)	0	7 (7.8)	1 (1.1)	1 (1.1)	0
Nonhematologic toxic effects								
Nausea and vomiting	77 (86.5)	12 (13.5)	0	0	76 (84.4)	13 (14.4)	1 (1.1)	0
Hepatotoxicity	8 (9.0)	2 (2.2)	1 (1.1)	0	7 (7.8)	4 (4.4)	1 (1.1)	0
Nephrotoxicity	2 (2.2)	3 (3.4)	0	0	5 (5.6)	0	0	0
Xerostomia	32 (36.0)	45 (50.6)	8 (9.0)	0	31 (34.4)	46 (51.1)	8 (9.0)	0
Auditory/hearing	32 (36.0)	25 (28.1)	1 (1.1)	0	37 (41.1)	19 (21.1)	6 (6.7)	0
Skin/neck tissue damage	53 (59.6)	8 (9.0)	0	0	48 (53.3)	3 (3.3)	0	0
Any late adverse events*	48 (53.9)	15 (16.9)	7 (7.9)	0	49 (54.4)	10 (11.1)	7 (7.8)	0
Dry mouth	44 (49.4)	6 (6.7)	1 (1.1)	0	45 (50.0)	4 (4.4)	1 (1.1)	0
Auditory/hearing loss	22 (24.7)	11 (12.4)	5 (5.6)	0	30 (33.3)	6 (6.7)	5 (5.6)	0
Peripheral neuropathy	11 (12.4)	0	0	0	17 (18.9)	0	0	0
Skin fibrosis	22 (24.7)	0	0	0	14 (15.6)	0	0	0
Dysphagia	8 (9.0)	0	0	0	15 (16.7)	1 (1.1)	0	0
Eye damage	1 (1.1)	0	1 (1.1)	0	1 (1.1)	0	1 (1.1)	0
brain injury	1 (1.1)	0	0	0	7 (7.7)	0	0	0

Abbreviations: UTI=Ulinastatin.

*Acute adverse events were graded according to the Common Terminology Criteria for Adverse Events (version 5.0). †Late adverse events were graded according to the Late Radiation Morbidity Scoring Scheme of the Radiation Therapy Oncology Group.

References:

1. Monti CB, Ambrogi F, Sardanelli F. Sample size calculation for data reliability and diagnostic performance: a go-to review. Eur Radiol Exp. 2024 Jul 5;8(1):79. doi: 10.1186/s41747-024-00474-w.
2. You R, Liu YP, Xie YL, Lin C, Duan CY, Chen DP, Pan Y, Qi B, Zou X, Guo L, Cao JY, Zhang YN, Wang ZQ, Liu YL, Ouyang YF, Wen K, Yang Q, Xie RQ, Li

HF, Duan XT, Ding X, Peng L, Chen SY, Liang JL, Feng ZK, Xia TL, Xie RL, Jiang R, Gu CM, Liu RZ, Sun R, Yang X, Liu LZ, Ling L, Liu Q, Ng WT, Hua YJ, Huang PY, Chen MY. Hyperfractionation compared with standard fractionation in intensity-modulated radiotherapy for patients with locally advanced recurrent nasopharyngeal carcinoma: a multicentre, randomised, open-label, phase 3 trial. *Lancet*. 2023 Mar 18;401(10380):917-927. doi: 10.1016/S0140-6736(23)00269-6. Epub 2023 Feb 23.

3. Miao J, Wang L, Tan SH, Li JG, Yi J, Ong EHW, Tan LLY, Zhang Y, Gong X, Chen Q, Xiang YQ, Chen MY, Guo Y, Lv X, Xia WX, Tang L, Deng X, Guo X, Han F, Mai HQ, Chua MLK, Zhao C. Adjuvant Capecitabine Following Concurrent Chemoradiotherapy in Locoregionally Advanced Nasopharyngeal Carcinoma: A Randomized Clinical Trial. *JAMA Oncol*. 2022 Oct 13;8(12):1776–85. doi: 10.1001/jamaoncol.2022.4656. Epub ahead of print.

4. Sapra RL. Power and sample size estimation for interim analysis using PASS. *Current Medicine Research and Practice* 7(1): 24-28, 2017. doi. 10.1016/j.cmrp.2017.01.004.

Reviewer #2

Reviewer’s comments	Authors’ responses or change made	Page number
1. This trial enrolled patients with AJCC 8th edition stage III-IVA NPC from 2018-2021. Did patients receive induction chemotherapy? If so, please list the regimens and assess for balance between cohorts. Induction chemotherapy may result in OM, and pre-existing OM was an exclusion criterion for this trial.	Thank you for this comment. We totally agree that induction chemotherapy may result in OM. As stated in the "Procedures" section of the manuscript, all patients underwent concurrent chemoradiotherapy (CCRT) and did not receive induction chemotherapy, since CCRT was one of the standard treatment options at that time.^{1,2} Furthermore, the presence of OM prior to treatment was one of our exclusion criteria, as shown in Study design and participants section of the Methods.	Revised lines 384-386 on page 18

2. Page 17: Can the authors confirm that Ulinastatin was given three times daily via IV on every day of radiotherapy? How was this logistically completed? Did patients have indwelling ports or PICCs? What interval separated Ulinastatin administration? What were criteria for discontinuation of Ulinastatin? Can the authors please describe previously-reported potential side effects or toxicities of Ulinastatin in the introduction or discussion?	We appreciate your professional suggestion on this matter. In accordance with the treatment standards at the five hospitals participating in this clinical trial, all patients receiving chemotherapy had an indwelling PICC. It is convenient for Ulinastatin to be administered intravenously through the indwelling PICCs. Ulinastatin was administered three times daily, at 8 a.m., 2 p.m., and 8 p.m, with 6-hour intervals between doses. The criteria for discontinuation of Ulinastatin were :1. The patient refused Ulinastatin treatment; 2. Allergic reaction during treatment; 3. Intolerance of associated toxicity. We have described the potential side effects or toxicities of Ulinastatin in the "Methods" and "Discussion" sections of the manuscript, found in lines 299-302 and 459-461.	Revised lines 298-302 on page 14 and 459-461 on page 21
3. Page 17: Please expand upon the standard approach for managing RTOM. What standard interventions did these five centers typically use? What types of oral analgesics were typically used? What types of oral and/or nasal rinses were typically used? Were any other mucositis-modifying agents used (e.g., glutamine, light therapy, etc). What were typical criteria for feeding tube placement? Please describe the general approach to treating presumed infections with antibiotics, as need for antibiotics is not commonly observed when treating oropharyngeal carcinomas with cisplatin-based chemoradiotherapy.	We thank the reviewer for your expertise comments. The standard approach for managing RTOM is described in detail in the "Study Protocol" section of Supplementary information, as follows: 1. Strengthen oral care and education; 2. Oral treatment before radiotherapy: replace metal fillings with non-metal fillings, remove metal dentures and affected teeth according to the advice of dentists, etc.;; 3. In the presence of mild pain (NRS pain score 1-3) and erythema of oral mucosa, non-opioid painkillers should be used to relieve pain and local sprays should be used to promote mucosal repair. 4. In the presence of moderate pain (NRS pain score 4-6) that does not interfere with transoral feeding, spotty oral ulcers, or scattered mucosal leukoplakia, dietary modifications were made to avoid stimulating or hard foods, and local analgesia with 0.5%-1.0% procaine gargle, systemic analgesia with weak opioids, and aerosolised throat sprays (pramipexole, chymotrypsin, etc.) were administered. 5. In the presence of severe pain (NRS pain score 7-9) that interferes with oral intake, fused oral ulcers or fused mucosal leukoplakia, and bleeding due to minor trauma, liquid diet and intravenous	Page 21-22 in Supplementary information

nutrition should be supplemented as appropriate, local analgesia with 0.5%-1.0% procaine, systemic analgesia with strong opioids, and spray throat should be given. If there is a specific infection, anti-infection treatment should be given.

6. In the event of obvious spontaneous bleeding due to necrosis of oral mucosal tissue or life-threatening complications, radiotherapy and chemotherapy should be stopped, and total intravenous nutrition should be given, as well as the above-mentioned local pain relief, systemic pain relief, anti-infection and other comprehensive symptomatic supportive treatment.

The following standard interventions for RTOM were typically used in the five centers of this study:

1. All patients should have their teeth cleaned to remove caries before radiotherapy.
2. when patients develop grade 2 RTOM, ultrasonic nebulized throat spray was administered.
3. When patients develop oral infections, the use of relevant antibiotics or antifungals is decided based on the results of bacterial and fungal cultures and drug sensitization of throat swabs.
4. When patients develop grade 4 RTOM, radiotherapy and chemotherapy should be stopped, and symptomatic treatment and total intravenous nutrition should be given.
5. According to the pain condition, appropriate analgesic drugs should be given.

Our commonly used oral analgesics are shown in the table below.

Grading of pain	Pain medication	oral analgesics
Mild	Non-opioid analgesics	Aspirin, Ibuprofen, Celecoxib
Moderate	weak opioids	Paracetamol and Dihydrocodeine Tartrate Tablets, Tramadol
Severe	strong opioids	Oxycodone and Acetaminophen Tablets, Morphine, Oxycodone Hydrochloride Sustained-release Tablets

	For patients with nasopharyngeal cancer, we typically choose 0.9% saline solution and 3% sodium bicarbonate for nasal irrigation, as well as antibacterial mouthwash such as cetylpyridinium chloride for oral rinsing. We did not use mucositis regulators such as glutamine or light therapy. The typical criteria for feeding tube placement in nasopharyngeal carcinoma mainly include the following aspects: When patients have intact gastrointestinal function but are unable to receive oral nutritional supplementation due to anatomical or primary disease-related factors, tube feeding enteral nutrition should be the first choice. For short-term applications, nasogastric intubation can be employed, while long-term requirements necessitate the implementation of percutaneous endoscopic gastrostomy (PEG) or jejunostomy. When a patient develops oral infections, we decide whether to use the relevant antibiotic or antifungal based on the the results of bacterial and fungal cultures and drug sensitization of throat swabs.	
4. "RTOM can disrupt therapy and even drive patients to stop it altogether. It can also hasten the growth of remaining tumor cells, trigger metastasis or recurrence of the original tumor, and lower patients' long-term survival rates." These statements are not fully supported by high-level clinical evidence. RTOM certainly impairs quality of life and may lead to interruptions/discontinuation of therapy, but a direct link between RTOM and tumor growth is not established.	Thank you for your professional comment. We apologize for not expressing this point clearly, which may have led to misunderstanding. We have revised this sentence in the revised manuscript: "RTOM can impair quality of life and interfere with treatment, or even cause patients to discontinue treatment entirely, thereby seriously compromising the effectiveness of antitumor therapy."	Revised lines 256-258 on page 12

References:

1. Chen YP, Wang ZX, Chen L, et al. A Bayesian network metaanalysis comparing concurrent chemoradiotherapy followed by adjuvant chemotherapy, concurrent chemoradiotherapy alone and radiotherapy alone in patients with locoregionally advanced nasopharyngeal carcinoma. *Ann Oncol* 2015; 26: 205–11.
2. Yan M, Kumachev A, Siu LL, Chan KK. Chemoradiotherapy regimens for locoregionally advanced nasopharyngeal carcinoma: a Bayesian network meta-analysis. *Eur J Cancer* 2015; 51: 1570–79.

Authors' response

Reviewer #1

Reviewer's comments	Authors' responses or change made	Page number
1. You successfully addressed all my suggestions. I was especially pleased that you were able to add information on late effects. Congratulations.	Thank you very much for the positive comment. Your expertise has profoundly improved the quality of our study, and we sincerely appreciate your dedication to refining this research.	----
2. Your response to my earlier comment: "3. Please define "cumulative incidence" of grade ≥ 3 toxicities. How does it differ from incidence?" implied that cumulative rate (as you now label it) at 7 weeks should equal incidence. In Figure 2 this is true for the control but not for the UTI group (26.97 vs. 25.8). Why are these figures different?	We sincerely appreciate your insightful comments. We apologize for the confusion caused by the incorrect figure inserted in the original manuscript. As we have defined, the cumulative rate of graded ≥ 3 acute RTOM at 7 weeks should be equal to the incidence rate. During our rigorous data verification process, we identified a discrepancy in the original dataset: one patient in the UTI group was erroneously recorded as having grade 3 RTOM from 4 to 7 weeks of radiotherapy. This error was promptly corrected in the revised dataset. However, due to an oversight during manuscript submission, the outdated version of the figure was mistakenly retained, resulting in an overestimation of the cumulative incidence of grade 3 RTOM by one case in the specified timeframe. Importantly, this revision does not alter the statistical significance of the findings or the overall conclusions of the study. The updated figure has been thoroughly validated and is included in the revised manuscript (please refer to Figure 2B in the updated version). The relevant data in the figure are also available in the source data uploaded along with the revised manuscript.	Figure 2B on page 37

Reviewer #2

Reviewer's comments	Authors' responses or change made	Page number
1. In response to Reviewer #2 Comment #1, I recommend explicitly stating on page 18 that no patient received induction chemotherapy.	Thank you for your suggestion. We have accordingly added the following to the methods section of the manuscript: No patient received induction chemotherapy.	Revised line 391 on page 18
2. In response to Reviewer #2 Comment #2, I recommend explicitly stating in the manuscript that a) patients received IV Ulinastatin at 8am, 2pm, and 8pm on every day of radiotherapy, b) the standard for chemotherapy in this region is that all patients have an indwelling PICC, and c) the criteria for discontinuation of Ulinastatin as stated by the authors in their response.	We appreciate your professional suggestion on this matter. We have added the following clarifications to the methods section of the manuscript based on your recommendations:  1. Lines 393-395: The standard for chemotherapy in participating treatment centers is that all patients have an indwelling Peripherally Inserted Central Catheter (PICC). 2. Lines 397-399: Ulinastatin was administered at 8 a.m., 2 p.m., and 8 p.m, with 6-hour intervals between doses. 3. Lines 403-405: The criteria for discontinuing Ulinastatin were as follows : 1. The patient refuse Ulinastatin treatment; 2. Allergic reaction during treatment; 3. Intolerance of associated toxicity. 	Revised lines 393-395 on page 18, 397-399 and 403-405 on page 19
3. In response to Reviewer #2 Comment #3, it is stated that intravenous nutrition was administered. Was enteral nutrition (nasogastric, gastrostomy) not performed for mucositis limiting oral intake? I recommend explicitly stating that glutamine and light therapy were not used in this trial.	Thank you for this comment. In this study, enteral nutrition via nasogastric tube or gastrostomy was not routinely administered for patients requiring nutritional support due to the following considerations: First, Exophytic nasopharyngeal tumors increase the technical difficulty of nasogastric tube placement and raise the procedural risks of iatrogenic injury. Second, gastrostomy is an invasive procedure that some Chinese patients are reluctant to accept due to cultural and personal preferences. In contrast, they are more likely to prefer parenteral nutrition support. Therefore, we encourage patients to supplement their nutrition with oral nutritional supplements (ONS) under adequate pain management. For those who cannot meet their nutritional needs through ONS, we assess the feasibility of nasogastric tube placement or gastrostomy. If patients decline these options, parenteral nutrition is considered.	Revised line 402 on page 19

	According to your suggestion, we have stated in methods section of the revised manuscript that glutamine and light therapy were not used in this trial.	
--	---	--